# The impact of thermal and auditory unpleasant stimulus on explicit motor imagery in healthy individuals: An experimental study

**Gabriel Cohen-Aknine** [1,2*], **Raphaël Pionnier** [2], **Denis Mottet** [1], **Arnaud François Dupeyron** [1,2]

**1** Euromov Digital Health in Motion, University of Montpellier, IMT Mines Ales, Montpellier, France,
**2** Service de Médecine Physique et de Réadaptation, CHU Nîmes, University Montpellier, Nîmes, France

\* gabriel.cohenaknine@chu-nimes.fr

## Abstract

Motor imagery is the ability to mentally simulate a motor task without actually performing it. Furthermore, pain is an unpleasant sensory experience that involves different dimensions – sensory-discriminative, motivational-affective, and cognitive-evaluative – that are known to interfere with motor imagery. However, it remains unclear which specific pain dimension most significantly impairs motor imagery vividness. This study aims to compare the effects of unpleasant auditory (primarily affective and cognitive) and thermal (primarily sensory) stimuli, which can be assimilated to pain, on discrete and continuous explicit motor imagery sessions. Eighteen healthy participants were exposed to these unpleasant stimuli in addition to a control condition. Participants rated the vividness of their explicit motor imagery after performing full back-and-forth flexion and extension of their wrists in discrete and continuous sessions. Results showed that during discrete explicit motor imagery, only the aversive auditory stimulus significantly reduced motor imagery vividness, whereas thermal pain had no effect. In contrast, motor imagery vividness was preserved during the continuous session. These findings suggest that explicit motor imagery may be more affected by the affective dimension of pain induced by aversive auditory stimuli. The preservation of motor imagery vividness in the continuous session provides insight into the optimization of rehabilitation programs.

## Introduction

Pain is defined by the International Association for the Study of Pain (IASP) as "an unpleasant sensory and emotional experience associated with actual or potential tissue damage, or described in terms of such damage" [1]. It is a multidimensional and dynamic process involving sensory-discriminative (e.g., intensity, localization), motivational-affective (e.g., anxiety, escape strategies) and cognitive-evaluative (e.g., attention, memory) dimensions [2–4].

provided the original author and source are credited.

**Data availability statement:** All relevant data are within the manuscript and its Supporting information files.

**Funding:** The author(s) received no specific funding for this work.

**Competing interests:** The authors have declared that no competing interests exist.

Exercise is known to reduce pain sensation, a phenomenon referred to as exercise-induced hypoalgesia [5]. However, in individuals complaining from chronic pain, the hypoalgesic effects of rehabilitation exercise are often mitigated [6–8]. As a result, alternative rehabilitation approaches such as motor imagery (MI) have been proposed. MI has been shown to be beneficial in rehabilitation [9,10], particularly for patients with pain [11,12], as it allows stimulation of the sensorimotor cortex without the pain associated with actual movement. However, Moseley et al. [13] showed an increase in pain intensity during motor imagery training for patients with chronic pain complains, such as complex regional pain syndrome. This finding may mitigate the promising results of this therapy.

MI is the ability to imagine performing a motor task without actually doing it [14]. It can take several forms, including implicit MI, which involves mentally rotating and recognizing images of the body [15,16], and explicit MI, which involves imagining the movement of a body part without actually performing the movement. Motor imagery activates brain regions that use similar substrates to those involved in sensorimotor or motor functions [17–19].

However, some studies suggest that the experience of pain may reduce implicit [20] and explicit MI vividness [21] in individuals with chronic pain, although these findings vary depending on the type of pain [22]. In experimental pain research, pain can be induced by various stimuli, such as thermal pain [23,24]. Based on the IASP definition of pain as an unpleasant experience, pain-like experiences have been observed with aversive auditory stimuli that produce pain intensities comparable to thermal pain and activate common brain patterns associated with negative valence and affective states [25–27]. A previous study showed that experimental pain could interfere with MI processes in healthy individuals [28]. In this study, the authors used a phasic pain stimulus involving electrical stimulation of the fifth digit and the knee during a thumb flexion/extension motor imagery task. The results showed a reduction in corticospinal excitability during motor imagery tasks regardless of location of the pain stimulus. However, to our knowledge, few information is available about whether the type of pain and its different dimensions (affective, cognitive and sensory) influence MI vividness.

In addition, the characteristics of the movement required to the participant can be disruptive. Movements can generally be performed in discrete (a single full or limited range of motion) or continuous (repeated full or limited range of motion) sessions [29]. Sensorimotor brain patterns are differentially activated by discrete and continuous movements [30,31]. Similarly, sensorimotor activity has been shown to vary between discrete and continuous MI sessions [32]. However, the effects of experimental unpleasant stimuli on discrete versus continuous MI tasks remain poorly documented and understood.

The aim of this study was to compare the effects of unpleasant thermal and auditory stimuli with a control resting state condition on explicit motor imagery vividness in healthy subjects during discrete and continuous sessions. Based on previous findings, we hypothesized that MI vividness would be reduced under unpleasant conditions regardless of the MI session (discrete or continuous).

## Materials and methods

This work is part of a larger, experimental single center prospective design study conducted at the Research Unit of Euromov Digital Health & Motion (Montpellier, France). The study was approved by the local ethics committee (Comité d'Éthique de la Recherche de l'Université de Montpellier: n° UM 2023−031) according to the declaration of Helsinski revised in 2013. Participants received and signed a written informed consent form.

### Participants

Eighteen healthy right-handed participants, aged 18–50 years, with no history of neurological or psychiatric disorders, no chronic pain and no pain in the upper limbs during the experiment were included. Students of Montpellier University were recruited to participate. Recruitment was carried out by email and posters around the university. Participants were excluded from the study if they were unable to perform the procedure described below and/or because of a dysfunction of any equipment used in the protocol or in case of adverse events. The inclusion period has been realized between November 20, 2023 and February 22, 2024. No sample size calculation was provided. However, some authors recommend a sample size of 10–20 subjects for similar studies [33].

Age, sex, weight, height, and body mass index (BMI) were recorded. Participants' handedness was assessed using the Edinburgh Handedness Inventory – Short Form (EHI-SF) [34], a questionnaire that scores handedness based on 10 daily activities, with a ratio and a cut-off of 40 points or more indicating right-handedness. Levels of physical activity were measured using the International Physical Activity Questionnaire-Short Form (IPAQ-SF) [35]. This self-report questionnaire uses an algorithm to calculate the number of minutes of physical activity (PA) per week, categorized into three levels (low, moderate and vigorous), and a score for inactivity (corresponding to the number of minutes spent sitting). The total score is expressed in MET (Metabolic Equivalent of Task) minutes/week defined as the ratio of the energy expended during the activity in question to the amount of energy expended at rest. For example, 1 MET corresponds to resting metabolic rate or basal metabolic rate [36].

Participants' explicit motor imagery vividness was assessed using the Kinesthetic and Visual Imagery Questionnaire (KVIQ) [37,38]. This questionnaire consists of 14 items: 7 for visual motor imagery (VMI), i.e., imagining the visualization of movements and 7 for kinesthetic motor imagery (KMI), i.e., imagining the sensation of movements. Participants were asked to perform different movements, before to imagine them and rate their motor imagery vividness on a 5-point Likert scale (1 = no image/no sensation to 5 = image as clear as seeing/as intense as performing the action).

### Procedure

Participants were seated in a chair in front of a computer in a relaxed position with their forearms resting on the armrests and their elbows naturally flexed in pronation, so that the wrists were fully flexed and relaxed. The procedure was divided in into two sessions: a discrete session consisting of single executed followed by imagined wrist movements, and a continuous session consisting of repeated executed followed by imagined wrist movements. All participants began with the discrete session, followed by the continuous session. Since all participants were right-handed (Table 1), the movement was executed and imagined using their dominant right hand against gravity due to the action of the extensor muscles. The instructions were displayed on a screen using PsychoPy software (version 2023.1.2) [39,40].

In both sessions, all participants performed "motor execution" followed by "motor imagery" tasks. During the motor execution and imagery tasks, a blank screen was displayed between the on-screen 'GO' and 'STOP' instructions. Before each task, general instructions were displayed to remind participants of the requirements (Fig 1).

Prior to the experiment, a training session was conducted to familiarize participants with the movement, consisting in performing a full range of motion wrist back-and-forth movement at a regular and consistent speed for four seconds, without acceleration or deceleration. Based on the kinesiology of the movement [41], the full range of motion of the wrist is

**Table 1. Descriptive characteristics of the participants.**

| Variable | Median | IQR | CI upper | CI lower |
|---|---|---|---|---|
| Age *(years)* | 23 | 3.5 | 27.117 | 20.383 |
| Size *(cm)* | 177.5 | 11.5 | 185.313 | 163.187 |
| Mass *(Kg)* | 69.5 | 19 | 87.778 | 51.222 |
| BMI *(Kg.m$^{-2}$)* | 22.55 | 4.15 | 26.492 | 18.508 |
| IPAQ-SF *(MET)* | 5003.5 | 3633.75 | 9426.957 | 2435.793 |
| IPAQ-SF Inactivity *(hours)* | 2310 | 1260 | 3522.09 | 1097.91 |
| KVIQ-V | 19 | 2 | 20.924 | 17.076 |
| KVIQ-K | 18 | 2 | 19.924 | 16.076 |
| Volume *(dB)* | 81 | 8 | 87.446 | 72.054 |
| Temperature *(°C)* | 46 | 1.875 | 47.741 | 44.1338 |
| EHI-SF | 80 | 25 | 111.884 | 63.1158 |

BMI, Body Mass Index. IPAQ-SF, International Physical Activity Questionnaire-Short Form. KVIQ-V: Kinesthetic and Visual Imagery Questionnaire - Visual score. KVIQ-K: Kinesthetic and Visual Imagery Questionnaire - Kinesthetic score. EHI, Edinburgh Handedness Inventory - Short Form. IQR, Inter-Quartile Range. CI: Confidence interval.

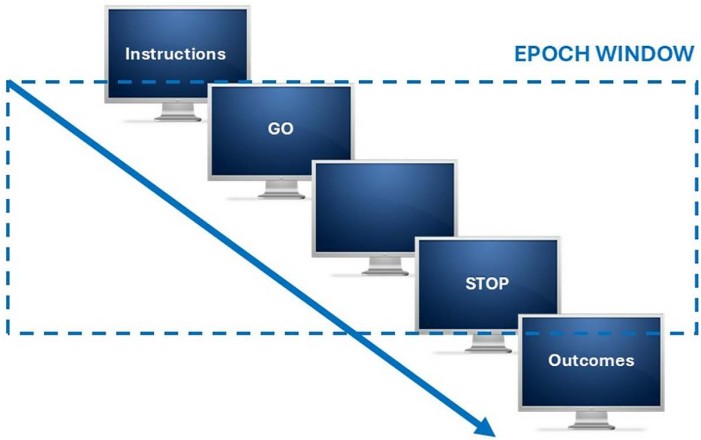

**Fig 1. Instructions design.**

120°, with back-and-forth joint movement; for a total of 240° in four seconds. Therefore, we can estimate that the movement has an arbitrary expected speed of 60° per second.

The instructions for the discrete and continuous sessions are described in parts 1 and 2 below.

**Part 1: discrete session.** With their wrists fully flexed, as described above, the participants were instructed to perform a single, full-range-of-motion wrist movement in the sagittal plane (from a fully flexed position to a fully extended position) and then return to the starting position within a 4-second time window, consistent with the training session. This procedure is known as the discrete session. The session began with motor execution, followed by a motor imagery task. During the motor imagery task, participants had to imagine performing the same movement. This procedure was repeated 20 times for each task (Fig 2).

**Part 2: continuous session.** Participants were instructed to perform repeated, 4-second, full-range-of-motion, back-and-forth wrist movements in the sagittal plane within a 25-second time window, consistent with the training session. They

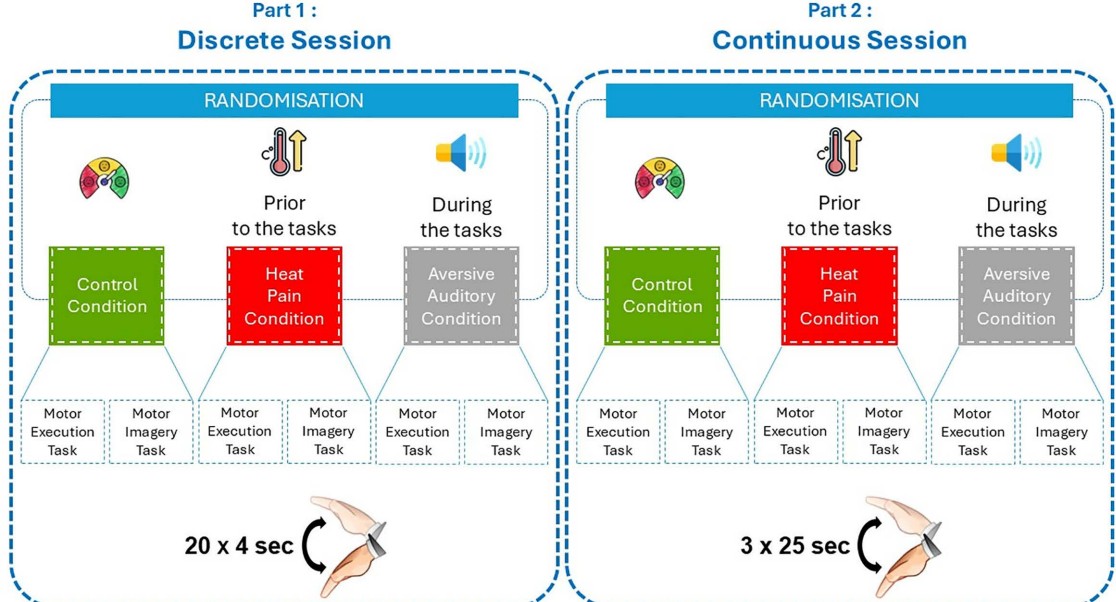

**Fig 2. Protocol design.**

started in the same position as in part 1. This procedure is known as the continuous session. Participants were instructed to repeat the four-second movement cycle until the word "STOP" appeared on the screen. The session began with motor execution, followed by a motor imagery task. During the motor imagery task, participants had to imagine performing the same movements within the same time frame. This procedure was repeated three times for each task (Fig 2).

The two unpleasant conditions and control condition were tested for both parts.

## Conditions

Three blocks of conditions were created: "control' (= no stimulus), 'auditory' (aversive auditory stimulus) and 'heat' (thermal painful stimulus). Our three-group comparison, which focused solely on unpleasant stimuli and excluded a pleasant group, was justified for two main reasons. Firstly, this approach aligns with previous studies that examined the impact of affective states (both pleasant and unpleasant) on the vividness of explicit motor imagery [42,43]. Secondly, our study focuses on negative affective states related to painful experiences, as defined by the IASP [1]. The order of the conditions was randomized using a dice, leading to six task sequence designs. In the "control" condition, movement was executed and imagined in the absence of aversive auditory or painful thermal stimuli (see S1 Fig).

The aversive 'auditory' stimulus was created using Audacity® (v 3.3.3) with a duration of 4 seconds, a start and end fade of 0.2 seconds, a frequency of 5000 Hz (sawtooth waveform) and a baseline volume of 75 dB. This stimulus was inspired by a previous publication by Valentini et al [26]. The thermal 'heat' stimulus was applied using hot water in a 14/19 L thermostatic bath (CORIO C-BT19, Julabo®, Seelbach, Germany) with a baseline temperature of 45°C [44]. The initial duration of the hot water immersion was based on a study by Granot et al. [44], in which participants experienced a 60-second immersion. However, due to the length of our protocol and to minimize the risk of adverse events, we reduced the immersion time to 30 seconds before the tasks began.

To standardize variability between participants, each participant selected their level of discomfort for each stimulus on a 100-point visual analogue scale (VAS) (0 = 'no discomfort' to 100 = 'unbearable discomfort'). Participants were instructed to

choose the water temperature and the auditory sound within a range between 60 and 75 on the VAS (Fig 3). The levels were adjusted in increments of 1°C for temperature and 1dB for sound volume until the chosen level of discomfort was reached.

Finally, the participants were asked to report any adverse events related to the study protocol, whether immediate or delayed.

For the thermal stimulus, the dominant hand was immersed in water up to the wrist joint before the task, and for the auditory stimulus, the sound was played through loudspeakers for 25-second periods during the task (Fig 2). Participants were stopped at mid-movement to ensure that the movement was completed without acceleration or deceleration at the end of the continuous session. The temperature level was monitored using the CORIO C-BT19 thermometer and the aversive auditory level was measured using the "Decibel X - Pro Sonomètre smartphone application" (SkyPaw Co.®, Ltd) [45]. A thermal heat stimulus was chosen because it results in a shorter duration of pain, a more constant quantitative description and a steeper slope for the intensity of the sensation compared to cold pain stimulus [46].

## Outcome measures

The effect of conditions on explicit motor imagery vividness was assessed using KVIQ's 5-point Likert scale. The two kinesthetic (KMI) and visual (VMI) motor imagery subscales were recorded for each session (discrete and continuous), and at the end of each condition (control, heat and auditory). The total score was calculated as the mean of the two subscales.

Additionally, the level of unpleasantness was assessed after each condition and task using the same Visual Analogue Scale (VAS), which that was established at the beginning of the protocol to measure self-rated levels of stimulus discomfort.

**Offline analysis.** Scale and subscale results were stored as Excel (Microsoft 365®) files using PsychoPy (version 2023.1.2) software and then analyzed offline.

## Statistical analysis

Statistical analyses were performed using R Studio® software (v. 2024.04.2). Due to the small sample [33,47], the Friedman test was used to compare conditions, and post hoc analysis was performed using the paired Wilcoxon test with the Bonferroni-Holm correction to adjust for multiple comparisons. For cases with missing data, the Skillings-Mack test was used instead of the Friedman test [48]. Outcome data were reported as median (interquartile range) and confidence intervals with upper and lower limits.

Effect sizes for statistically significant differences between conditions were calculated using Kendall's tau. Statistics are presented with a 95% confidence interval and an alpha risk of 0.05.

## Results

### Demographic analysis

Eighteen participants (7 females and 11 males) were included in the study. According to the EHI laterality scores, all participants were right-handed.

The descriptive analysis is detailed in the Table 1.

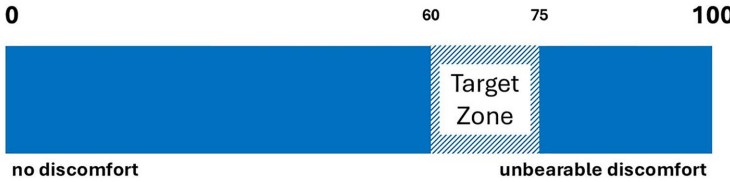

**Fig 3. Analogue visual scale for unpleasant stimulus levels.**

## Explicit motor imagery vividness

All results are presented in S1 Table.

### Discrete motor imagery. KVIQ total score

There were significant statistical differences between conditions (p=0.002). Post-hoc analysis revealed that the auditory stimulus decreased explicit motor imagery vividness compared to the control condition (p=0.001) with a large effect size (d=0.703) (Fig 4). However, there were no significant differences between the auditory and heat conditions or between the heat and control conditions.

### KVIQ Kinesthetic score

There were significant statistical differences between conditions (p=0.005). Post-hoc analysis revealed that the auditory stimulus decreased explicit kinesthetic motor imagery vividness compared to the control condition (p=0.037) with a large effect size (d=0.605), and compared to the heat condition (p=0.045) with a large effect size (d=0.601) (Fig 4).

### KVIQ Visual score

There were significant statistical differences between conditions (p=0.007). Post-hoc analysis revealed that the auditory stimulus decreased explicit visual motor imagery vividness compared to the control condition (p=0.021) with a large effect size (d=0.699) (Fig 4).

## Continuous motor imagery

**Total KVIQ score.** There were no significant statistical differences between conditions (p=0.645), indicating that the unpleasant conditions did not impair explicit motor imagery vividness during the continuous motor imagery task.

**KVIQ kinesthetic score.** There were no significant statistical differences between conditions (p=0.663), indicating that the unpleasant conditions did not impair explicit kinesthetic motor imagery during the continuous motor imagery task.

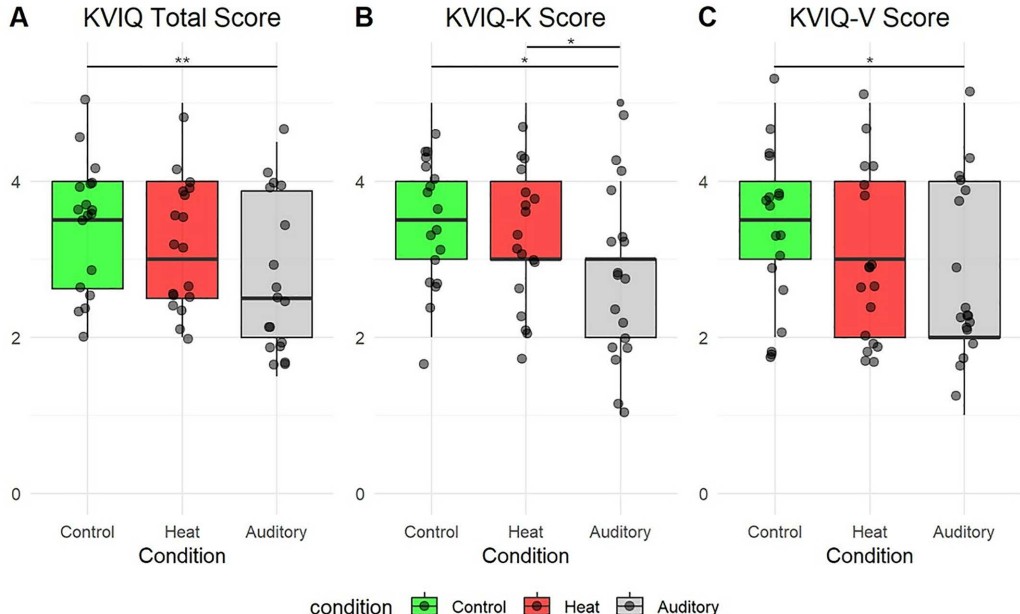

**Fig 4. Box plot of KVIQ scores between Control, Aversive Auditory and Thermal Heat conditions during discrete motor imagery. A; KVIQ Total score. B, KVIQ Kinesthetic Score. C, KVIQ Visual Score.**

**KVIQ visual score.** There were no significant statistical differences between conditions (p = 0.103), indicating that the unpleasant conditions did not impair explicit visual motor imagery during the continuous motor imagery task.

**Levels of unpleasantness.** Unpleasant VAS scores were significantly higher in aversive and heat conditions than in the control condition, in all sessions and for all tasks (p < 0.001) with large effect sizes (between 0.777 and 0.885). Post hoc analysis revealed that the auditory aversive stimulus led to higher VAS scores for discrete motor imagery than the heat pain stimulus (p = 0.008) with a large effect size (d = 0.672) (Fig 5).

A statistical analysis of all motor imagery and unpleasantness scores is provided in the S1 File.

Throughout the study, none of the participants reported any adverse events, whether during the discrete or continuous session.

## Discussion

Our hypothesis that explicit motor imagery vividness would be reduced by unpleasant stimuli was partially confirmed, as only the aversive auditory stimulus decreased explicit MI scores, specifically during discrete explicit motor imagery session. Indeed, for the continuous motor imagery session, our hypothesis was not confirmed for either unpleasant stimulus.

Firstly, consistent with previous study [49], we found a decrease in explicit motor imagery vividness during the aversive auditory stimulus, but only during the discrete session, not the continuous one. Some authors have postulated that motor imagery may be more influenced by cognitive-evaluative or motivational-emotional factors [50]. In our study, we used two unpleasant stimuli to differentiate distinct aspects of pain, contrasting a stimulus without a localized component (aversive auditory stimulus) with a conventional experimental pain model (thermal heat stimulus), which provides a precise and unilateral localization related to the sensory-discriminative dimension of pain. The aversive auditory stimulus is likely more related to the cognitive-evaluative or motivational-emotional dimension due to its generally unpleasant characteristics and the global nature of the stimulus. In contrast, the thermal heat stimulus integrates the sensory-discriminative dimension, which is related to more localized unpleasant characteristics. On the one hand, with respect to the cognitive-evaluative

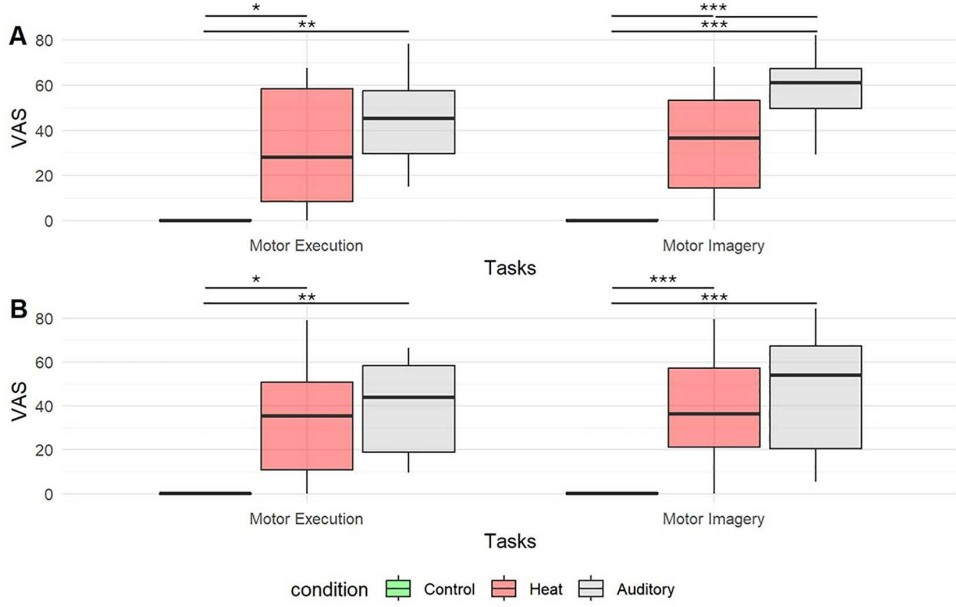

**Fig 5. Box plot of unpleasant VAS scores between Control, Aversive Auditory and Thermal Heat conditions during discrete (A) and continuous sessions (B).** *, p < 0.05. **, p < 0.01, ***p < 0.001.

dimension, previous research has shown that explicit motor imagery elicits high cognitive demand [12,50]. In addition, studies have shown that motor imagery relies on working memory storage and induces mental fatigue [51] whether during prolonged implicit [52] or explicit motor imagery [53]. Furthermore, research has shown a relationship between working memory and the perception of aversive sensations in response to unpleasant stimuli [54,55].

On the other hand, regarding the motivational-emotional dimension, however, some researchers have compared aversive auditory stimuli to a tinnitus-like sensation, which has been shown to activate cortical areas involved in attention and negative emotional responses [56]. Studies have also shown that aversive stimuli, such as auditory stimulus, elicit amygdala responses [57] related to negative emotions, such as fear and anxiety [58]. Furthermore, aversive auditory stimuli have been associated with conditioned threats and observed changes in alpha brain oscillations [59]. Unpleasant and threatening situations are involved in motivational behaviors, as described in fear-avoidance models [60,61]. Finally, a recent neurophysiological review proposed two brain pathways: one related to thalamo-somatosensory tracts involved in sensory-discriminative processing and another related to thalamo-prefrontal and cingulate tracts involved in cognitive-evaluative or motivational-emotional processing [62]. Our results suggest that the cognitive-evaluative and motivational-emotional dimensions of pain may be more related to impair explicit motor imagery vividness, when assessed using an auditory stimulus. However, these proposals should be approached with caution given the one-day experiment of the motor imagery tasks in the protocol of our study.

These disturbances were observed only in the discrete motor imagery session, not in the continuous. It has been suggested that brain activity in cortical motor planning areas increases during discrete versus continuous motor sessions [30], with greater attentional demands in the discrete session [29]. In support of this, previous research has shown that continuous movements are less affected by cognitive load than discrete movements [31]. In our study, the discrete session consisted of 60 trials (3 × 20 tasks) of discrete explicit motor imagery, whereas the continuous session consisted of 9 trials (3 × 3 tasks) of continuous explicit motor imagery. The high cognitive demands associated with explicit motor imagery, particularly in the discrete session, combined with the impact of the aversive auditory stimulus on the attentional network, may explain our findings. This may make the discrete motor imagery task more sensitive to the affective-motivational and cognitive-evaluative dimensions of pain.

Regarding the lack of effects of the thermal heat pain stimulus on explicit motor imagery vividness, our results contrast with previous studies suggesting that experimental pain impairs the motor imagery process [49]. However, different experimental pain models have been used in the literature, including phasic pain (short duration with intense sensation) and tonic pain (long duration with moderate sensation). For example, a previous study reported a decrease in cortico-spinal excitability during electrical pain stimulation while participants performed explicit motor imagery of digit V [49], whereas our experimental pain model used a thermal heat pain stimulus. Although thermal and electrical stimuli target nociceptors, research suggests that different experimental pain models engage multiple dimensions of the participant's experience, with no significant correlation observed between responses to electrical and thermal pain stimuli [23]. This suggests that experimental pain models may activate different sensory and affective pathways, resulting in unique individual experiences.

Some reviews have highlighted the impact of pain conditions on implicit motor imagery performance, showing a decrease in accuracy and response times [20,63], regardless of pain localization. Accordingly, some authors have suggested that changes in motor imagery are specifically related to a painful body part [64] likely due to changes in internal body representation [65]. However, other studies have shown that patients with widespread pain, such as fibromyalgia, have reduced implicit motor imagery vividness [66,67], suggesting a more complex interaction between pain localization and motor imagery deficits. Additionally, a study including women with menstrual pain demonstrated a reduction in explicit motor imagery vividness, assessed with the same questionnaire as in our study, with results showing a negative correlation between non-localized unpleasant sensory, such as somatic complaints related to menstrual symptoms, and motor imagery vividness scores [68]. In our study, we used an aversive auditory stimulus, a non-localized unpleasant sensory input, and found

that explicit motor imagery performance was more significantly impaired than that observed with a localized thermal unpleasant stimulus. These results suggest that the localization of the unpleasant stimulus alone does not fully explain explicit motor imagery deficits. Indeed, another study highlight the role of the affective dimension in motor imagery performance, reinforcing the idea that motivational-affective, and cognitive-evaluative factors interact to influence motor imagery process [43], and suggesting that these dimensions are also related to the general aspect of pain, irrespective of its localization.

The intensity of the unpleasant sensation after induced stimulus was at least equivalent, if not higher than, that in the aversive auditory condition compared to the control and heat conditions, whether during discrete and continuous motor imagery. These results are consistent with previous studies [26,69]. Finally, a complementary correlation analysis was performed and showed a negative moderate to strong correlation between unpleasantness intensity and explicit motor imagery vividness for the aversive auditory stimulus but not the thermal heat pain stimulus, confirming our findings regarding the nondisruptive effect of localized stimuli on explicit motor imagery vividness (complete analysis is provided in the Appendix 2). This suggests that the affective and cognitive dimension of pain is more likely to decrease explicit motor imagery process.

### Clinical implications

Our results showed that the continuous motor imagery was not impaired by unpleasant stimuli. These findings support the use of explicit motor imagery as a valuable therapeutic tool, as self-reported motor imagery vividness appear to be preserved in the presence of localized pain. This highlights an opportunity to optimize the implementation of motor imagery exercises and reinforces its potential as a rehabilitation strategy, given its demonstrated efficacy in reducing pain and improving range of motion [70]. Therefore, personalized explicit motor imagery therapy is recommended according to rehabilitation goals and continuous sessions may be considered, based on our results. However, our results were assessed in an experimental design study, thus, longitudinal evaluation should be conducted to provide a more robust therapeutic approach. Additionally, our results suggest that the affective and cognitive dimensions of pain, as measured by an aversive auditory stimulus, impair the vividness of explicit motor imagery specifically in discrete sessions. Previous studies have shown that psychosocial aspects of pain influence quality of life in interdisciplinary pain rehabilitation [71]. Thus, assessment of the affective dimension of pain [72] may be important prior to initiating rehabilitation programs, particularly those involving motor imagery in a discrete session. Previous authors have suggested that significant dosage or duration of motor imagery training is required to effectively decrease pain intensity [73,11], suggesting caution in interpreting the results.

### Limitations

First, our results were based on a small sample size, which may have limited our ability to capture the influence of gender, a factor known to influence the temporal aspects of explicit motor imagery [74]. However, a previous factor analysis of several experimental pain responses indicated that gender did not influence thermal heat pain and that 20% of the variance in pressure pain threshold was explained by heat pain sensitivity, supporting the reliability of our findings [23]. In addition, our sample consisted of relatively young participants with a median age of 23 years. As age has been shown to influence motor imagery vividness [74–76], this may have introduced a potential bias in our results.

Another potential limitation of our study is that the participants had high levels of physical activity (15 of 18 participants exceeded 3000 MET/min/week, classified as vigorous physical activity), which may not be representative of the general population. A meta-analysis has shown that athletes have better tolerance to experimental pain [77] and that individuals with high motor expertise, such as athletes, rely on different neural mechanisms during motor imagery and are considered as good imagers [78].

Third, the literature includes a wide variety of motor imagery assessment tools, ranging from clinical measures such as the Lateral Judgment Task for implicit motor imagery [79], and mental chronometry [80] to self-report questionnaires for explicit motor imagery [81], as well as neuroimaging techniques such as electroencephalography [82] and functional

magnetic resonance imaging [83]. This methodological diversity may have limited our conclusions and requires caution in generalizing our findings.

Finally, expectations of positive or negative experiences, commonly referred to as placebo or nocebo effects, suggest that an individual's predictive cognition of unpleasant stimuli could introduce bias into our study [84]. However, previous research has shown that volunteers participating in experimental pain studies tend to be less sensitive to pain and exhibit less fear of pain. This may mitigate potential bias, as any observed differences in individuals with lower pain sensitivity suggest that the effect size may be even larger in the general population.

## Opportunities

Recent research on corticospinal excitability suggests that the effects of experimental pain differ from those of clinical pain, highlighting the need for new experimental pain models to better understand how clinical pain affects motor bevahior [85]. In particular, aversive auditory stimuli may provide an opportunity to disrupt attentional processes in ways similar to those observed in patients with chronic pain [86]. Furthermore, while experimental pain models have been shown to replicate the spatial distribution of pain, they often fail to capture the qualitative aspects of pain as assessed by tools such as the McGill Pain Questionnaire [87]. Therefore, combining aversive auditory stimuli with thermal heat pain may provide a more comprehensive model that incorporates additional affective and cognitive dimensions of pain [23].

## Conclusion

This study shows that explicit motor imagery vividness in healthy individuals is differentially affected by the type of unpleasant stimulus and task. The motivational-affective and cognitive-evaluative dimensions of pain are more likely to impair discrete motor imagery compared to thermal pain, despite similar levels of unpleasantness. However, when performed in a continuous session, motor imagery vividness is preserved. These findings suggest that continuous explicit motor imagery may be a more effective therapeutic approach, although attention must be paid to fatigue and working memory disturbances. Future research could explore the combination of aversive auditory and conventional pain models in a longitudinal design.

## Supporting information

**S1 Fig. Randomization design.**
(DOCX)

**S1 Table. Summary of the KVIQ vividness total, kinesthetics and visual scores in each condition.**
(DOCX)

**S1 File. Statistical analysis.**
(PDF)

## Acknowledgments

We would like to thank the 2 research engineers Pierre Jean and Simon Plat from Euromov DHM for their help with software installation, logistics and signal analysis.

## Author contributions

**Conceptualization:** Gabriel Cohen-Aknine.

**Data curation:** Raphaël Pionnier, Denis Mottet.

**Formal analysis:** Gabriel Cohen-Aknine, Raphaël Pionnier.

**Investigation:** Gabriel Cohen-Aknine.

**Methodology:** Gabriel Cohen-Aknine, Denis Mottet, Arnaud François Dupeyron.

**Project administration:** Denis Mottet, Arnaud François Dupeyron.

**Supervision:** Denis Mottet, Arnaud François Dupeyron.

**Validation:** Raphaël Pionnier, Denis Mottet, Arnaud François Dupeyron.

**Visualization:** Denis Mottet, Arnaud François Dupeyron.

**Writing – original draft:** Gabriel Cohen-Aknine.

**Writing – review & editing:** Raphaël Pionnier, Denis Mottet, Arnaud François Dupeyron.

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
