## [Editor Report · Decision Letter 0]

11 Mar 2025

Dear Dr. COHEN-AKNINE,

Thank you for submitting your manuscript to PLOS ONE. After careful consideration, we feel that it has merit but does not fully meet PLOS ONE’s publication criteria as it currently stands. Therefore, we invite you to submit a revised version of the manuscript that addresses the points raised during the review process.

**ACADEMIC EDITOR:** in order to facilitate reviewing, could you please :

1. provide the checklist on which your essay is based (consort? strobe? spirit?)

2. modify the title to explicitly mention the method?

We look forward to receiving your revised manuscript.

Kind regards,

Thomas Rulleau, PT PhD

Academic Editor

PLOS ONE

Additional Editor Comments:

dear colleague,

in order to facilitate reviewing, could you please :

1. provide the checklist on which your essay is based (consort? strobe? spirit?)

2. modify the title to explicitly mention the method?

---

## [Author Response · Author response to Decision Letter 1]

16 Mar 2025

Dear ACADEMIC EDITOR, Thomas Rulleau,

Please take note of the changes made to the manuscript, highlighted in blue, in response to your suggestions. Our answers to your questions have been written in blue in the manuscript and below:

1. Please indicate the checklist on which your paper is based (CONSORT? STROBE? SPIRIT?).

After reviewing the EQUATOR website, we found the CRED-nf checklist to be the most appropriate for our experimental research design.

2. Change the title to explicitly mention the method.

As per your suggestion, we have revised the title to include "a controlled experimental study".

We thank you for your consideration.

Sincerely,

---

## [Decision Letter · Decision Letter 1]

22 Jun 2025

Dear Dr. COHEN-AKNINE,

We look forward to receiving your revised manuscript.

Kind regards,

Thomas Rulleau, PT PhD

Academic Editor

PLOS ONE

**Additional Editor Comments:**

dear colleague,

several points were raised by the reviewers and I invite you to respond. The most important point is to support and explain how we can be sure that it is the intervention that explains the evolution. thank you.

Reviewers' comments:

Reviewer's Responses to Questions

**Comments to the Author**

Reviewer #1: (No Response)

Reviewer #2: (No Response)

2. Is the manuscript technically sound, and do the data support the conclusions?

Reviewer #1: (No Response)

Reviewer #2: Yes

3. Has the statistical analysis been performed appropriately and rigorously?

Reviewer #1: (No Response)

Reviewer #2: Yes

4. Have the authors made all data underlying the findings in their manuscript fully available?

Reviewer #1: (No Response)

Reviewer #2: Yes

5. Is the manuscript presented in an intelligible fashion and written in standard English?

Reviewer #1: (No Response)

Reviewer #2: Yes

Reviewer #1: Reviewing PONE-D-25-10295_R1-1

I would thank the Editor, Doctor Thomas Rulleau for his invitation to perform this review of the manuscript entitled “The impact of thermal and auditory unpleasant stimulus on motor imagery in healthy individuals: an experimental study » by Cohen-Aknine Gabriel, Pionnier Raphaël, Pr Mottet Denis and Pr Dupeyron Arnaud.

I would like to thank the Authors for submitting their article. I acknowledge that a previous revision was performed. Based on the revised version of the manuscript, I formulated comments structured in 3 parts : majors, additionals and comments on the text that could further improve the quality of the manuscript.

Major comments:

- Regarding the conditions of the exposure. Is there any chance that pleasant stimuli would have resulted in MI improvement ? This is not discussed whereas it could have been. The rational of using the 3 groups (and not more) could be elaborated in the manuscript.

- Result part should be revised with not corresponding text and Table. This discrepency result in a text hard to follow and understand.

- Title could include an explicite mention of the pilot or proof-of-concept study rather or in extension to experimental study.

In line with this comment, please mention that no sample size was computed.

- Reading of the text and figures don’t explicitely inform about the movement speed between the continuous and discrete wrist flexion – extensions. Please clarify whether the movement speed was similar and controlled between conditions ;

Additional comments:

- In the title, abstract and the rest of the manuscript, it could be explicitely state that the authors refers to explicit motor imagery. Also, please use MI vividness instead of ability.

- Please elaborate the outcome of the measure since impairing motor imagery is not engouht precise.

- It seems to be essential to dinstiguish different aspects separating those refering to the delay in experiencing disease such as pain (and inducing potentially long term plasticity changes) to those elicited by MI training during a single session (i.e., more likely associated to short term changes).

- Some details could be added regarding the instruction provided to the participant during MI to complete the control of potential experimental bias.

- Did the authors face adverse effect with the use of painful thermal condition ? Please reporte this in the manuscript (in the result part) as long as the procedure to manage such event (in the method).

- In the outcomes, authors state using of the Likert scale of the KVIQ and report result of the MIQ which is confusing and should be corrected.

Pleasr also report the unpleasant mesure of the condition in the method to describe the procedure and report results in the corresponding part.

Comment on the text.

In the intro, ref 11, 12 about the rational of using MI without pain. Moseley et al., have shown that in case of complex regional pain syndrome, a non-negligible (~50%) of participants did showed an increase in pain intensity. Please elaborate.

Line 70: did the authors referes to implicit or explicit MI ? Please elaborate.

Line 77-79 Results of Ref 31 could be presented here.

Line 81-88 : The rational of comparing discrete vs continuous MI could be elaborate. The authors could find some information in the recent paper by : Suica, Z., Behrendt, F., Gäumann, S., Gerth, U., Schmidt-Trucksäss, A., Ettlin, T., & Schuster-Amft, C. (2022). Imagery ability assessments : A cross-disciplinary systematic review and quality evaluation of psychometric properties. BMC Medicine, 20(1), 166. https://doi.org/10.1186/s12916-022-02295-3 Also, please revise the term modality which refers to visual or kinaesthetic and seems not appropriate here. In this part, the clinical implication of this work could be added.

Participant, please elaborate the choice of the KVIQ as MI questionnaire vs other questionnaires validated in French (i.e., MIQ-3 2020, 10.1051/sm/2019035).

Procedure : please revise the word modality which is inappropriately used.

A Figure could be helpful to better understand the experimental procedure.

Line 150-1 : there was no specific instructions regarding visual or kinaesthetic motor imagery modalities for [MI]. The term modality is used appropriately here. The question is rather about the control of the MI : how was it perform in this condition ?

Part 1 – 2. In addition to a new figure, this part is quite hard to follow with motor execution and imagery introduced before part 1. Please revise this section. It appears that there is figures for instruction design and protocol. These figures appears after the text and may be introduced earlier.

Part 1. Regarding the position, it appears that the wrist flexion extension is performed against gravity (i.e., similarly to a MRC manual muscle testing score = 3/5). Is this correct ? If so, please add this information in the text.

Part 2. Did the participant received an instruction for movement speed ? Please add this information in the text.

Conditions : please elaborate the rational for choosing each of them. Hot exposition seems to be presented lines 197 – 200. Is there recommended duration needed for exposure to hot water for the occurrence of pain ? Please elaborate.

It could be more appropriate to start with detailing each condition and then specifying the order. Regarding the order, it is not clear how the dice was used to generate the random order in the conditions. Please elaborate.

Line 174. The authors state “without unpleasant stimulus” which seems to refers to only auditory one but not to painful one. I suggest a revision of this sentence.

Primary outcome :

Is it only the effect of pain sensation which is examined ? I would suggest a revision as there are 3 different conditions. This could include the effect of conditions on MI vividness change (see my next comment).

I also recommend a word changing MI ability into MI vividness using the Likert score of the KVIQ.

Statistics: Is there no additional exploratory analyse that can be performed relating the activity (IPAQ) and vividness score and separating the effect of gender ? Or regarding the MI vividness and an unpleasant measure of score (if any was measured) ? This latter would have been of particular interest for the present study. In addition, is there any differences regarding the MI ability. Are poor imagers more affected by conditions as compared to good imagers ?

Results :

245. I recommend moving at the end of the result the mention of the appendix 1.

249. Is the significant difference reported only for the auditory condition vs the 2 others ? please elaborate. Table 3 (continuous condition) appears twice first in the discret condition and again in the continuous one. Understanding of the results is were hard with the presentation of the table. Please revise the result part for the main outcome.

250. In the method, the authors refers to KVIQ and here to MIQ-RS. Please clarify what Likert scale was used to measure the MI vividness.

258. There seems to be a typo “).” Before “Summaries”.

274. There seems to be an error in the reference.

Discussion.

See major comment.

Also, is there a chance that specifying instruction would have reduced the negative effect of the unpleasant stimulus ? Alternatively, the use of a pleasant or rhythmic auditory stimulus ?

The authors differentiate the cognitive and affective aspect of pain and explain their results (decrease of MI vividness after auditory condition). But the auditory condition could alter MI vividness because of an increase in the cognitive load while maintaining engaged in MI vividness. Please elaborate why the results are related to affective aspect more than cognitive on ?

311. Please revise this statement since only a single session would have been performed.

364-6 this could be summarized in the result part.

366-9 this could be introduced earlier in the manuscript in the method and results. It could be added in the Table 1.

385-7. I would rather temper this statement. On one hand, the result is obtained from a pilot study. On the other hand, the choice of the exercises (either continuous or intermittent) should be adapted to rehabilitation goal. It seems relevant to imagine continuous movement for walking but much less for balance exercises or grasping movement.

Figures 4 and 5. I recommend to add the individual points of the 18 participants in the boxplots.

I wisk good luck to the author for the revision of their manuscript.

Yours sincerely,

Dr Sébastien Mateo.

Reviewer #2: I thank the editor for inviting me to review this manuscript.

This manuscript presents a controlled experimental study investigating the impact of unpleasant stimuli on motor imagery abilities in a healthy population. The originality of this study lies in testing two different pain induction paradigms: an aversive auditory stimulus, which targets the affective component of pain and is fairly representative of the sensations experienced in chronic pain conditions, and a classical thermal stimulus, commonly used in the majority of laboratory pain induction protocols.

The study is clear and well-structured. The introduction provides an overview of recent literature in the field, the manuscript is easy to follow. The objective and hypotheses are defined and well-articulated. Although the work is fundamentally theoretical, its clinical implications are significant and thoroughly discussed. The limitations of this work are also clearly stated.

There are several minor comments to address, notably to the methods section.

Comment 1 Total score. I understand the methodological choice not to analyze the imagery modalities (visual and kinesthetic) separately, and to compute a total score, given the absence of specific instructions on how to perform the imagery. However, I question the relevance of adding the scores from the two subscales to obtain a total score out of 10, rather than calculating an average, which would have kept the results within the original 1-to-5 scale of the KVIQ. This would have made score interpretation easier by maintaining consistency with the original structure of the scale. Indeed, median scores around 4.5 remain difficult to interpret: it is not entirely clear whether the movement was "well" imagined, even though the reported imagery abilities suggest good quality imagery.

Comment 2. Modality of motor imagery instructions. In the Methods section, it is stated that participants were explicitly instructed to use both visual and kinesthetic motor imagery. However, did the authors ask participants which modality they actually used to imagine the movement? Furthermore, since a total score (VMI + KMI) was calculated, do we know which modality was preferentially used by the participants?

Lines 106-107 → "Students and staff of the University of Montpellier were recruited to participate." Did the participants have any prior experience with motor imagery?

Comment 3. Regarding the wrist movements, with which hand are they performed physically and/or imagined? It seems to me that this is not specified in your protocol. I assumed they were done with the left hand (figure 3)—could you please confirm?

Comment 4 The thermal stimulus. This brings me to another question regarding the thermal stimulus. The right hand was immersed in water up to the wrist joint — but for how long exactly? Was it immersed throughout the entire duration of the discrete modality (4 seconds × 20) and the continuous modality (3 × 25 seconds)? Was the hand removed from the water between blocks?"

Comment 5. Auditory stimulus. I also have a similar question regarding the auditory stimulus. It is stated that 'the sound was played for periods of 25 seconds through loudspeakers' — but when exactly did this occur? Was it during each block of the continuous modality, and was it repeated or paused between blocks?

I believe the authors should consider adding a legend to Figure 3 to clarify these details that appear to be missing, which are important for fully understanding the protocol.

Comment 5. Results. There is an issue with the formatting regarding the results and the numbering of the tables. I noticed that the results for the discrete motor imagery modality appear twice: first between lines 254 and 255 (Table 2: normal median = 6.5; heat median = 6.0; auditory median = 4.5), and again between lines 280 and 281. I believe this is a duplication error. Please kindly correct the formatting, possibly by placing the tables at the end of the manuscript.

If I’m not mistaken, here is the proposed numbering for the tables:

- Table 1: Descriptive characteristics of the participants

- Table 2: Summary of the Total MIQ-RS score in each condition during the discrete motor imagery modality

- Table 3: Summary of the Total MIQ-RS score in each condition during the continuous motor imagery modality

Furthermore, you state that you used the KVIQ scales as the primary outcome measure. However, in the manuscript, the tables are labeled with "Total MIQ-RS" scores. This inconsistency is confusing. Could you please clarify this point: did you use the KVIQ or the MIQ-RS?

And, please correct the reference error on line 275: (Error! Reference source not found).

**Do you want your identity to be public for this peer review?** For information about this choice, including consent withdrawal, please see our Privacy Policy

Reviewer #1: **Yes: ** Sébastien MATEO

Reviewer #2: No

---

## [Author Response · Author response to Decision Letter 2]

14 Aug 2025

RESPONSE TO REVIEWERS

We sincerely thank the editors and the two reviewers for their insightful and constructive suggestions, which have significantly contributed to improving the quality of our manuscript.

Please note that we have provided both a Clean Copy and a revised manuscript, with all comments and changes highlighted in blue. Our responses to the reviewers are also included in this letter and are marked in blue for clarity.

Reviewers' comments:

Reviewer's Responses to Questions

Comments to the Author

1. If the authors have adequately addressed your comments raised in a previous round of review and you feel that this manuscript is now acceptable for publication, you may indicate that here to bypass the “Comments to the Author” section, enter your conflict of interest statement in the “Confidential to Editor” section, and submit your "Accept" recommendation.

Reviewer #1: (No Response)

Reviewer #2: (No Response)

2. Is the manuscript technically sound, and do the data support the conclusions?

Reviewer #1: (No Response)

Reviewer #2: Yes

3. Has the statistical analysis been performed appropriately and rigorously?

Reviewer #1: (No Response)

Reviewer #2: Yes

4. Have the authors made all data underlying the findings in their manuscript fully available?

Reviewer #1: (No Response)

Reviewer #2: Yes

5. Is the manuscript presented in an intelligible fashion and written in standard English?

Reviewer #1: (No Response)

Reviewer #2: Yes

6. Review Comments to the Author

Reviewer #1: Reviewing PONE-D-25-10295_R1-1

I would thank the Editor, Doctor Thomas Rulleau for his invitation to perform this review of the manuscript entitled “The impact of thermal and auditory unpleasant stimulus on motor imagery in healthy individuals: an experimental study » by Cohen-Aknine Gabriel, Pionnier Raphaël, Pr Mottet Denis and Pr Dupeyron Arnaud.

I would like to thank the Authors for submitting their article. I acknowledge that a previous revision was performed. Based on the revised version of the manuscript, I formulated comments structured in 3 parts : majors, additionals and comments on the text that could further improve the quality of the manuscript.

Major comments:

- Regarding the conditions of the exposure. Is there any chance that pleasant stimuli would have resulted in MI improvement ? This is not discussed whereas it could have been. The rational of using the 3 groups (and not more) could be elaborated in the manuscript.

We would like to thank Reviewer 1, Dr Sébastien Mateo, for his suggestion. Our selection of three groups was based on our objective of focusing on unpleasant sensory stimuli and their impact on motor imagery vividness, particularly in relation to clinical pain and rehabilitation applications. Other researchers have indeed described the effects of pleasant, neutral and unpleasant stimuli on motor imagery vividness. For example, Bywaters et al. (10.1080/09658210444000160) have demonstrated that affective states involving pleasant and unpleasant imagery modify motor imagery vividness. These findings were further supported by a study by Wriessnegger et al. (10.1016/j.bandc.2018.07.006), which used functional near-infrared spectroscopy to detect increased cortical activity during negative, unpleasant imagery related to painful experiences. Consequently, we have added a rationale section to the 'Materials and Methods' explaining our choice of the three groups, as follows:

Line 201-206: ”Our three-group comparison, which focused solely on unpleasant stimuli and excluded a pleasant group, was justified for two main reasons. Firstly, this approach aligns with previous studies that examined the impact of affective states (both pleasant and unpleasant) on the vividness of explicit motor imagery (46,47). Secondly, our study focuses on negative affective states related to painful experiences, as defined by the IASP (1).”

- Result part should be revised with not corresponding text and Table. This discrepency result in a text hard to follow and understand.

Thank you for your feedback. Following your suggestion, we have revised the Results section and added a new table, Table S1, in Appendix 1. We apologize for the error regarding the questionnaire name, which was mistakenly referenced from another study. The necessary corrections have been made in line with the reviewer's suggestions.

The results are now presented in two distinct parts: one for the discrete sessions and one for the continuous sessions. These results account for the total scores, as well as the kinesthetic and visual modalities separately. For greater clarity, all the results are summarized in Table S1, which has been included in Appendix 1.

- Title could include an explicite mention of the pilot or proof-of-concept study rather or in extension to experimental study.

In line with this comment, please mention that no sample size was computed.

Thank you for these suggestions. Our decision not to calculate the sample size was based on previous experimental studies focused on pain models, such as those involving thermal painful stimulation. The following reviews reference these studies: Bank et al. (10.1002/j.1532-2149.2012.00186.x), Devecchi et al. (10.1097/J.PAIN.0000000000002819), and Izadi et al. (10.3389/fnhum.2022.863741). However, some authors have reported guidelines related to sample size calculation, such as Serdar et al. (10.11613/BM.2021.010502). For experimental studies, these authors recommend a sample size of 10 to 20 subjects for group comparisons. The rationale for the absence of a sample size calculation has been added to the "Materials and Methods" section as follows:

Line 109-121: “No sample size calculation was provided. However, some authors recommend a sample size of 10 to 20 subjects for similar studies (37).”

- Reading of the text and figures don’t explicitely inform about the movement speed between the continuous and discrete wrist flexion – extensions. Please clarify whether the movement speed was similar and controlled between conditions ;

We agree with this important point regarding the movement speed performed by the participants. Additional details have been added to the "Materials and Methods" section as follows:

Line 157-163: “Prior to the experiment, a training session was conducted to familiarize participants with the movement, consisting in performing a full range of motion wrist back-and-forth movement at a regular and consistent speed for four seconds, without acceleration or deceleration. Based on the kinesiology of the movement (45), the full range of motion of the wrist is 120°, with back-and-forth joint movement; for a total of 240° in four seconds. Therefore, we can estimate that the movement has an arbitrary expected speed of 60° per second.

Additional comments:

- In the title, abstract and the rest of the manuscript, it could be explicitely state that the authors refers to explicit motor imagery. Also, please use MI vividness instead of ability.

Thank for this indication. Additional precisions about the type of motor imagery and the use of “vividness” were used to replace the term “modality”, in accordance with the suggestion and the following article : https://doi.org/10.1016/j.neures.2018.12.005

In addition, the title was changed as suggested:

Line 1-3:” The impact of thermal and auditory unpleasant stimulus on explicit motor imagery in healthy individuals: an experimental study”

- Please elaborate the outcome of the measure since impairing motor imagery is not engouht precise.

We only used the two KVIQ 5-point Likert subscales for kinesthetic and visual motor imagery to measure motor imagery vividness. We did not use other tools. All detailed results are provided in Appendix 1 Table S1.

- It seems to be essential to dinstiguish different aspects separating those refering to the delay in experiencing disease such as pain (and inducing potentially long term plasticity changes) to those elicited by MI training during a single session (i.e., more likely associated to short term changes).

Thank you for this significant suggestion. Additional information has been added to the "Clinical Implications" section to clarify the distinction between the short- and long-term effects of motor imagery training. According to Suso-Martí et al. and Cuenca-Martinez et al., motor imagery training may require repetition and a prolonged duration to decrease pain intensity. The following sentence was added in the Discussion section:

Line 391-393:” However, these proposals should be approached with caution given the one-day experiment of the motor imagery tasks in the protocol of our study.

Additionally, the sentence below was added to the Clinical Implications section:

Line 469-471:” Previous authors have suggested that significant dosage or duration of motor imagery training is required to effectively decrease pain intensity (77,78), suggesting caution in interpreting the results.”

- Some details could be added regarding the instruction provided to the participant during MI to complete the control of potential experimental bias.

We have rewritten the instructions and description of the procedure for greater readability.

Line 157-165:” Prior to the experiment, a training session was conducted to familiarize participants with the movement, consisting in performing a full range of motion wrist back-and-forth movement at a regular and consistent speed for four seconds, without acceleration or deceleration. Based on the kinesiology of the movement (45), the full range of motion of the wrist is 120°, with back-and-forth joint movement; for a total of 240° in four seconds. Therefore, we can estimate that the movement has an arbitrary expected speed of 60° per second.

The instructions for the discrete and continuous sessions are described in parts 1 and 2 below.

And line 170-189:

“Part 1: discrete session

With their wrists fully flexed, as described above, the participants were instructed to perform a single, full-range-of-motion wrist movement in the sagittal plane (from a fully flexed position to a fully extended position) and then return to the starting position within a 4-second time window, consistent with the training session. This procedure is known as the discrete session. The session began with motor execution, followed by a motor imagery task. During the motor imagery task, participants had to imagine performing the same movement. This procedure was repeated 20 times for each task (Fig. 2).

Part 2: continuous session

Participants were instructed to perform repeated, 4-second, full-range-of-motion, back-and-forth wrist movements in the sagittal plane within a 25-second time window, consistent with the training session. They started in the same position as in part 1. This procedure is known as the continuous session. Participants were instructed to repeat the four-second movement cycle until the word "STOP" appeared on the screen. The session began with motor execution, followed by a motor imagery task. During the motor imagery task, participants had to imagine performing the same movements within the same time frame. This procedure was repeated three times for each task (Fig. 2).”

- Did the authors face adverse effect with the use of painful thermal condition ? Please reporte this in the manuscript (in the result part) as long as the procedure to manage such event (in the method).

Thank you for this important suggestion. Of course, all participants were closely supervised during each session to prevent any adverse effects. If any adverse events were detected, the procedure was immediately stopped. To date, no participants have reported any adverse events.

In accordance with Reviewer 1’s suggestion, we have added the following sentences to the "Material and Methods" section:

Line 116-118:” Participants were excluded from the study if they were unable to perform the procedure described below and/or because of a dysfunction of any equipment used in the protocol or in case of adverse events.

and line 213-217:” The initial duration of the hot water immersion was based on a study by Granot et al. (48), in which participants experienced a 60-second immersion. However, due to the length of our protocol and to minimize the risk of adverse events, we reduced the immersion time to 30 seconds before the tasks began.”

Line 224-225:” Finally, the participants were asked to report any adverse events related to the study protocol, whether immediate or delayed.”

Additionally, the absence of adverse events was reported in the "Results" section:

Line 347-348:” Throughout the study, none of the participants reported any adverse events, whether during the discrete or continuous session.”

- In the outcomes, authors state using of the Likert scale of the KVIQ and report result of the MIQ which is confusing and should be corrected.

Pleasr also report the unpleasant mesure of the condition in the method to describe the procedure and report results in the corresponding part.

We sincerely apologize for this oversight. The MIQ was indeed used in another study, and reusing the R codes caused this error. We have made the necessary corrections.

Comment on the text.

In the intro, ref 11, 12 about the rational of using MI without pain. Moseley et al., have shown that in case of complex regional pain syndrome, a non-negligible (~50%) of participants did showed an increase in pain intensity. Please elaborate.

Thank you for the opportunity to discuss the effectiveness of motor imagery and the need for caution regarding its immediate effects on specific populations. In response to the reviewer's comment, we revised the "Introduction" section to include findings from Moseley et al. 2008 (10.1002/art.23580), noting that about 50% of participants with complex regional pain syndrome experienced increased pain intensity. The following elaboration has been added to the "Introduction" section:

Line 65-68:” However, Moseley et al. (13) showed an increase in pain intensity during motor imagery training for patients with chronic pain complains, such as complex regional pain syndrome. This finding may mitigate the promising results of this therapy.”

Line 70: did the authors referes to implicit or explicit MI ? Please elaborate.

This explanation refers to both implicit and explicit MI. We have made the corresponding changes:

Line 75-77:” However, some studies suggest that the experience of pain may reduce implicit (24) and explicit MI vividness (25) in individuals with chronic pain, although these findings vary depending on the type of pain (26).”

Line 77-79 Results of

---

## [Editor Report · Decision Letter 2]

31 Aug 2025

The impact of thermal and auditory unpleasant stimulus on explicit motor imagery in healthy individuals: an experimental study

PONE-D-25-10295R2

Dear Dr. COHEN-AKNINE,

We’re pleased to inform you that your manuscript has been judged scientifically suitable for publication and will be formally accepted for publication once it meets all outstanding technical requirements.

Kind regards,

Hesam Ramezanzade, Ph.D

Academic Editor

PLOS ONE
---

## [Editor Report · Acceptance letter]

PONE-D-25-10295R2

PLOS ONE

Dear Dr. COHEN-AKNINE,

I'm pleased to inform you that your manuscript has been deemed suitable for publication in PLOS ONE. Congratulations! Your manuscript is now being handed over to our production team.

Kind regards,

on behalf of

Dr. Hesam Ramezanzade

Academic Editor

PLOS ONE